# Notes on the Taxonomy of *Salix vitellina* (Salicaceae)

**DOI:** 10.3390/plants12142610

**Published:** 2023-07-11

**Authors:** Alexander M. Marchenko, Yulia A. Kuzovkina

**Affiliations:** 1Russian Park of Water Gardens, Polevaya Str., 12, Klyazma mkr, 141230 Pushkino, Russia; 2Department of Plant Science and Landscape Architecture, University of Connecticut, 1376 Storrs Rd., Storrs, CT 06269, USA

**Keywords:** Linnaeus’s specimen, *S. alba*, *S*. × *fragilis* f. *vitellina*, *S. euxina*, willow

## Abstract

*Salix vitellina* L., or golden willow, was described by C. Linnaeus in 1753. It was later considered to be affiliated with *S. alba*, and its taxonomic rank has been changed to variety, subspecies, and form. A recent proposal designated it as a form of *S. alba* × *S. fragilis*. The goal of this study was to verify the taxonomic designation of *S. vitellina* using morphological characteristics including ovule number. A few specimens of *S. vitellina* from Europe and North America, including the lectotype LINN1158.13, were analyzed. It was recorded that *S. vitellina* has an ovule index of 6–10, with most valves with four and five ovules and less than 50% of valves with five ovules. These ovule parameters were similar to those of *S. alba*. The other floral characteristics also indicated that *S. vitellina* is associated with *S. alba*. No signs of androgyny or flower aberrations, commonly occurring in willow hybrids, were found in the specimens of *S. vitellina*. Thus, the analyses did not corroborate the hybrid origin of *S. vitellina*. The ovule analysis also confirmed that f. *chermesina* with orange–red stems is also a taxon of *S. alba*, which differs from f. *vitellina* by a greater ovule index of 12–16.

## 1. Introduction

*Salix* L. (Salicaceae), comprising approximately 450 species of woody plants [1], is a genus of considerable taxonomic complexity due to individual and phenotypic variation, dioecy, hybridization, introgression, and allopolyploidy. *Salix alba* and *S. fragilis*, which are closely related species characterized by the lanceolate, acuminate, and serrulate leaves, represent one of the most difficult taxonomic groups in the genus [2]. Many molecular studies have been conducted to clarify the definitions of these species, and some of them have shown high genetic similarity and a shared ancestry for *S. alba* and *S. fragilis* [3,4,5,6,7,8,9,10,11,12]. Yet, the boundaries between these species are defined by relatively few diagnostic characters, resulting in various taxonomic interpretations of their taxa, including the taxonomic affiliation of *S. vitellina* L.

*Salix vitellina*, or golden willow, was described by C. Linnaeus in 1753 [13]. The specific epithet “vitellina” translates from Latin as “egg-yolk yellow” or “dull yellow just turning red (Lindley)” [14] referring to the bright yellow or yellow with red and orange colors of the 1–3-year-old stems. *Salix vitellina* is common in cultivation around the world but has limited cold hardiness. Skvortsov [15] suggested that this taxon (as *S. alba* var. *vitellina*) is of southern origin because of its continuous seasonal growth, when cultivated in northern regions, and poor adaptation to low winter temperatures. Neumann [16] also suggested that *S. vitellina* (as *S. alba* subsp. *vitellina*) is a possible native species in the Balkan Peninsula (as var. *vitellina*) because it occurs as a wild plant along riversides in Croatia, while in Central Europe, it is only present as a cultivated plant.

There are many basket and ornamental cultivars of *S. vitellina*, which are cultivated around the world for various purposes. *Salix vitellina* is a parent, along with *S. babylonica,* of *S.* × *pendulina* ’Chrysocoma’, a commonly cultivated weeping willow with yellow stems [17].

### 1.1. Analysis of the Linnean Protologue

There are three synonyms in the protologue of *S. vitellina* [13]: “Salix foliis ferratis ovatis acutis glabris: ferraturis cartilagineis, petiolis callofo-punctatis. Hort.upf. 295”, “Salix foliis lineari-lanceolatis acuminatis. Gvett.ftamp. i. p. 206”, and “Salix sativa lutea, folio crenato. Bauh. pin. 473.” While the second and third synonyms most likely describe the same species with lanceolate serrulate leaves and correspond to the herbarium specimen LINN 1158.13, the first synonym likely defines a different taxon with ovate serrulate leaves. It is possible that Linnaeus observed these different taxa with bright yellow–orange stems and included them under the name of *S. vitellina.*

### 1.2. Nomenclature

*Salix vitellina* was typified on LINN 1158.13 [18]. First described as a species of *Salix vitellina* by C. Linnaeus in 1753, the taxonomic rank of this taxon was changed to a variety, subspecies, and form. It is a basionym for other names, including *S. alba* var. *vitellina* (L.) Stokes (1812), *S. alba* subsp. *vitellina* (L.) Schübl. and G.Martens (1834), *S. alba* f. *vitellina* (L.) Wimm. (1866), and the most recent *S.* × *fragilis* f. *vitellina* (L.) I.V.Belyaeva (2018) [19]. It should be noted that, previously, Belyaeva proposed to use *S.* × *fragilis* as a binomial for the hybrid of *S*. *alba* × *S. fragilis,* while replacing the traditional *S. fragilis* L. with a new name *S. euxina* I.V.Belyaeva. In our previous paper, we clarified that Belyaeva’s view on the nomenclature of *S. fragilis* and the application of the name *S. euxina* to the crack willow were erroneous and one should return to the classical nomenclature of *S*. *alba* L. and *S. fragilis* L. and their hybrid *S.* × *rubens* Schrank [20], which was used by Skvortsov [15] or Meikle [21]. 

The need for a better understanding of the taxonomic affiliation of *S. vitellina* was of interest to the authors, as we have recently advanced taxon analysis in *Salix* using the ovule number method. In *Salix,* the traditional morphological characteristics have a high degree of variability, which limits their diagnostic value. Relying on these characteristics for taxa delineation is not always a dependable approach especially in some complex taxonomic groups such as *S. alba* and *S. fragilis* [20,22]. Chmelař [23], Argus [1,24], and Valyagina-Malutina [25] were the first to recognize that in *Salix,* the number of ovules per ovary represents a fundamental phylogenetic characteristic. Our recent studies have demonstrated that, in addition to the traditional morphological characteristics, the number of ovules present in the ovaries of the willow flower is a useful diagnostic trait. Each willow species has a characteristic ovule index (the minimum and maximum number of ovules per ovary), as well as specific fractions of ovaries and valves with a different number of ovules. This information provides diagnostic clues in some cases of taxonomic uncertainties. Combined with traditional morphological characteristics, the ovule number can be used to support taxonomic decisions in various taxonomically challenging groups at various ranks in the genus *Salix* [20,22,26,27,28,29]. Recent studies have indicated that *S. alba* and *S. fragilis,* which are not easy to differentiate based on traditional morphological characteristics, are distinguishable by their ovule number [20,22]. Hence, a study of *S. vitellina* was conducted to confirm its taxonomic affiliation using the ovule number along with other floral characteristics.

## 2. Materials and Methods

### 2.1. Plant Materials

*S. vitellina*: Four specimens were analyzed based on their names which included the epithet “*vitellina*” at some taxonomic level. Specimen 1: The authentic herbarium specimen LINN 1158.13, which was designated as a lectotype for *S. vitellina* (Figure 1). Specimens 2–4: *S. vitellina* from various parts of Europe and North America. 

Related specimens: Specimens 5–12: A few specimens of *S. alba* without yellow stems, which were previously analyzed for another study and were determined to have an ovule index of 6–10 [20,22], were included in these analyses for comparison. Specimen 13: An artificial hybrid of *S. vitellina* called *S.* ‘Oranzhevaya Tolstostvol’naya’ and Specimen 14—one of its parents [22]. Specimen 15: S. alba ‘Nova’ from the Dendrological Garden in Průhonice (accession No. 2000.10493), Czech Republic, originally procured from Brochet-Lanvin nursery, France, and grown at the Russian Park of Water Gardens, Moscow, Russia, since 2011. 

### 2.2. Flower Morphology

The morphological analyses of the flowers were conducted using a Nikon SMZ800N stereomicroscope (Nikon, Tokyo, Japan).

### 2.3. Ovule Count

The ovule count was performed according to the previously described experimental protocol [26,27,28]. All normally developed ovaries of one catkin were examined for each specimen. The ovaries were opened along the median veins of the two profiles. The number of ovules on each of the two valves was recorded. There were valves with either the same or a different number of ovules in the same ovary. The ovule index was determined as the minimum–maximum number of ovules per ovary in a catkin (for example, 6–10). For each specimen, the fractions of ovaries with different numbers of ovules and the fractions of valves with different numbers of ovules were recorded. These data resulted in the detailed characterization of the genotypes, which, combined with the morphological characteristics, assisted in the selection of the homogeneous groups.

## 3. Results and Discussion

### 3.1. Flower Morphology

The structure of the pistillate flowers of the studied specimens of *S. vitellina* suggested their affiliation with *S. alba* (Figure 2 and Figure 3). The floral bracts of *S*. *alba* have glabrous tips and curled hairs at the base; the pure *S. fragilis* has straight long hairs at the tip and the base; the hybrids *S. alba* × *S. fragilis* usually combine both hair types [30]. The type specimen LINN 1158.13 had typical *S. alba* bracts (Figure 2).

Our previous microscopic analysis of the pistillate flowers revealed some important diagnostic characteristics of *S. alba* and *S. fragilis* [20]. For example, obclavate capsules and cushion-shaped nectaries are characteristic of *S. alba,* while pyriform capsules and cup-shaped nectaries occur in *S. fragilis*. The analysis of LINN 1158.13 and other specimens of *S. vitelllina* revealed their similarity to the pistillate flowers of *S. alba*, and more specifically to the flowers from the *S. alba*-*fragilis*—Group [22] based on the ratio and relative sizes of the flower bracts, nectaries, stigmas, and stipes to the other flower parts. In *Salix,* the ratio comparison is more critical for the identification than the absolute size of the organs [15]. In all specimens of *S. vitellina,* the obclavate capsules had a gradually tapering style and short stipe; slightly bifid stigmas had four lobes; floral bracts with abaxial side pubescent at the base were much longer than the stipes; and in all specimens, there was one adaxial cushion-shaped nectary, sometimes completely enclosing the stipe. The flower bracts of *S. vitelllina* lacked the long straight white trichomes, which in *S. fragilis* should exceed the bracts in length [31]. No signs of androgyny or flower aberrations, commonly occurring in willow hybrids, were found in specimens of *S. vitellina*. 

### 3.2. Ovule Number

The four specimens of *S. vitellina* had similar distributions of ovules and ovule indices (Table 1). Specimens 1–3 had an ovule index of 6–10, with most ovaries having 7–9 ovules. Specimen 4, for which only 2 ovaries were available for the analysis, also had an ovule index of 7–9. The limited number of the analyzed ovaries of this specimen suggests that, if more ovaries were available, its ovule index could also be 6–10.

Overall, this ovule index was within the ovule range for *S. alba* [20,22,23,24,25]. The distribution of ovules in Specimens 1–4 corresponded to the *S. alba*-*fragilis*—Group [22]. A few specimens (Specimens 5–12) belonging to this group were added to this study for comparison. They were characterized by having less than 50% of valves with three ovules, in parallel with the greater number of valves with four and five ovules, and the ovule index was 6–10, as in the specimens of *S. vitellina*. The name “*S. alba*-*fragilis*—Group” indicates an association with *S. fragilis* because of the presence of some valves with three ovules (three ovules per valve are characteristic of *S. fragilis*, which has the most valves, with three ovules and an ovule index of 6-6). 

The following average distributions were recorded for the specimens 5–12: 10% of ovaries had 6 ovules/ovary distributed as 3/3 ovules/valve (i.e., both valves had 3 ovules), 21 % of ovaries had 7 ovules/ovary distributed as 3/4 ovule/valve (i.e., there were valves with 3 and 4 ovules), 11 % of ovaries had 8 ovules/ovary distributed as 3/5 ovule/valve (i.e., there were valves with 3 and 5 ovules), 27 % of ovaries had 8 ovules/ovary distributed as 4/4 ovule/valve (all valves had 4 ovules), 22 % of ovaries had 9 ovules/ovary distributed as 4/5 ovule/valve (i.e., there were valves with 4 and 5 ovules), and 9 % of ovaries had 10 ovules/ovary distributed as 5/5 ovule/valve (all valves had 5 ovules). This ovule distribution was identical to *S. vitellina* (Figure 3M–O). 

Specimens 5–12 were initially identified by prominent salicilogists as belonging to either *S. alba, S. fragilis,* or *S. alba* × *S. fragilis*, indicating that all specimens had characteristics common to *S. alba* and *S. fragilis*, with the most frequent identification as *S. alba.* These specimens had the majority of valves with four and five ovules, which is typical for *S. alba*. They exhibited mostly the characteristics of this species. At the same time, the presence of some valves with three ovules resulted in the occurrence of some morphological characteristics of *S. fragilis,* prompting some researchers to designate them as *S. fragilis* or a hybrid *S. fragilis* × *S. alba.* We previously reported that while *S. alba* and *S. fragilis* are closely related species and that the boundaries between them are defined by relatively few diagnostic characteristics, these species can be distinguished by their ovule number [20]. *Salix fragilis* has the majority of valves with three ovules, while *S. alba* has the majority of valves with 4–5 and a larger number of ovules.

Thus, specimens 1–4 of *S. vitellina* had an identical ovule distribution to specimens 5–12 of *S. alba.* All specimens had most valves with four and five ovules (76% in Specimens 1–4 and 75% in Specimens 5–12) and some valves had three ovules (24.4% in specimens 1–4 and 24.0% in Specimens 5–12). Therefore, the ovule number analysis suggests that *S. vitellina* belongs to *S. alba*. The only difference between specimens 1–4 and specimens 5–12 was the stem color as was reflected in the epithet “vitellina” for Specimens 1–4.

### 3.3. Analysis of the Lectotype Specimen of S. vitellina

The lectotype LINN 1158.13 of *S. vitellina* [18] consists of two branches: one with female and another with male catkins (Figure 1). This lectotype was studied by Belyaeva et al. [19], and it was concluded: “Examination of the lectotype, “Herb. Linn. No. 1158.13, female specimen (LINN)” showed that the plant on this herbarium sheet had characteristics not only of *S. alba* L. but also of *S. euxina* I.V.Belyaeva and is, in fact, a hybrid between these two species”, for which the name is *S*. × *fragilis* L., described by Belyaeva [32], should be used.

Our analysis of the comparative characteristics of *S. vitellina*, *S. alba,* and *S. euxina* (formerly *S. fragilis*) presented by Belyaeva et al. [19] indicated that mostly overlapping morphological characteristics were listed for the three taxa, precluding their clear differentiation. Amongst them, the two characters demonstrated as evidence that *S. vitellina* has the characteristics of *S. euxina* were the presence of brittle branches and buds with blackening apices due to bud scale dieback. However, it was impossible to determine if the branches of LINN 1158.13 were fragile, as the specimen was not annotated with this information. Regarding the blackening apices of bud scales, there were no buds on the specimen LINN 1158.13, so it was impossible to record this characteristic. From our own analysis of various specimens of *S. vitellina* there were no buds with blackening apices. In addition, the flower bracts of *S. vitellina* were listed by Belyaeva et al. [19] as variable between *S. alba* and *S. euxina*; however, as mentioned above, our analysis of LINN 1158.13 revealed their similarity with *S. alba* (Figure 3). Thus, there was no evidence that *S. vitellina* has the characteristics of *S. euxina* and should thus be considered as a hybrid *S. × fragilis* (*S. alba* × *S. euxina*). Belyaeva et al. [19] speculated on the hybrid origin of *S. vitellina* and did not provide sufficient evidence to support this proposal.

Moreover, some observations by horticulturists noted that branches of cultivars of *S. vitellina* were not fragile, and the plants exhibited a clearly upright habit typical of *S. alba* but not a spherical crown typical of *S. fragilis* [33].

### 3.4. A Hybrid Study

Experimental hybridization of *S. vitellina* was conducted by Shaburov at the Botanic Garden of the Ural Branch of the Russian Academy of Sciences, Yekaterinburg, in 1960 [26,34,35,36,37]. *Salix vitellina* is not hardy in the Urals, and so, Shaburov made a cross to produce a cold-hardy hybrid progeny of golden willow.

Two plants—male and female representatives (listed by Shaburov as *S*. *alba*)—were collected by Shaburov in the same natural habitat in the Urals. They had similar upright crowns, with them moderately drooping after 10 years of age. The female specimen was included in this study (Specimen 14). The ovule analysis of Specimen 14 indicated that it belongs to the *S. fragilis-alba*—Group, which has most valves with three and some valves with four and five ovules and the ovule index of 6–9. The second parent that was used in this cross was *Salix vitellina* procured from Latvia. Two offspring—female *S*. ‘Oranzhevaya Tolstostvol’naya’ (meaning “An Orange Thick-Stemmed” in Russian) and male ‘Pamyati Bazhova’ (meaning “In Bazhov’s memory”)—were selected from this progeny [26,35,36] (https://osf.io/45tkq; accessed on 28 April 2023). *Salix* ‘Oranzhevaya Tolstostvol’naya’ represented a robust female hybrid up to 16 m in height with yellow stems. The stem color and structure of the floral branches of *S*. ‘Oranzhevaya Tolstostvol’naya’ were similar to *S. vitellina* (Figure 4D,E), but its elongated crown became slightly weepy after 10 years similar to its second parent (Figure 4A–C). Also, its extreme cold hardiness (this cultivar is cold hardy in the Urals and the Perm Krai with a continental climate, characterized by long, cold, and snowy winters) was also inherited from the second parent belonging to the *S. fragilis-alba*—Group. 

The ovule number in a willow hybrid is the statistical mean of the ovule number of its parents [23,26,27,28,29], and a hybrid with a known genetic background, such as *S*. ‘Oranzhevaya Tolstostvol’naya’, can be used for a “backward study” to elucidate its pedigree or to calculate the *predicted* ovule index for one of its parents. The analysis of *S*. ‘Oranzhevaya Tolstostvol’naya’ (Table 1; Specimen 13) revealed its ovule index of 6–10. Knowing the ovule number of one parent (Specimen 14), it was possible to predict the ovule number for the second parent *S. vitellina,* which was 6–9.5 or 6–10. Thus, the *predicted* ovule index corroborated the *calculated* ovule index for *S. vitellina* previously recorded for Specimens 1–4.

Importantly, it was documented that *S*. ‘Oranzhevaya Tolstostvol’naya’ had about 15% abnormal ovaries with three valves instead of two valves, typical for willow species. Flower aberrations commonly occur in willow hybrids. However, as stated above, there were no aberrations in Specimens 1–4, which, again, asserts that *S. vitellina* is not a hybrid. Another offspring from the same cross, which Shaburov called ‘Pamyati Bazhova’, was a profusely blooming male willow. It also produced an abnormal catkin with one pistillate flower, which had seven ovules arranged as three and four on each of the two valves (Figure 4G–I). This ovule distribution was within the ovule index of 6–10 for S. ‘Oranzhevaya Tolstostvol’naya’ confirming their relatedness and the accuracy of the ovule methodology.

### 3.5. Salix Alba Chermesina

The botanical references listed different taxa similar to *S. vitellina*, though their citations were accompanied by vague descriptions. Hartig [38] believed that there were two variants of *S. alba* with bright yellow stems: *S. alba* var. *vitellina* and *S. alba* var. *chermesina*. Späth [39] listed “*S. alba vitellina*”, “*S. alba vitellina* Britzensis”, and “*S. alba vitellina* Nova”. Rehder [40] listed “*S. alba* var. *vitellina*” and “*S. alba* f. *chermesina*” and considered the epithet ‘Britzensis’ as a synonym of *S. alba* f. *chermesina*. Stott [41] listed “*S. alba vitellina*” and “*S. alba chermesina*” and noted that a cultivated variety ‘Vitellina Nova’ belongs to *S. alba chermesina*. This assessment suggests the presence of at least two similar taxa with yellow–orange–red stems.

A notable specimen procured from Prühonice under the name *S. alba* ‘Nova’ (specimen 15) was grown and has been observed by the authors in the field since 2011 (Figure 5). This plant had an orange–red color of the 1–3-year-old branchlets rather than the yellow color of *S. vitellina*. *Salix alba* ‘Nova’ was not fully hardy in Zone 4, where it was grown as a coppiced specimen, as its branchlets were damaged by cold winter temperatures. The branchlets of ‘Nova’ were scarcely pubescent with patched hair. The buds were glabrous, brightly colored as branchlets with a pronounced keel and live scales. After opening, the buds’ scales remain attached to the stems for a long time. Its leaves were thick, oval-elongated (rather than narrow lanceolate as in *S. vitellina*), pointed, and serrated. They were weakly pubescent on both sides, with more pronounced pubescence above the veins. The stipules were small, triangular with glands, and persistent.

‘Nova’ was a female plant, and its ovule number was recorded (it was not included in Table 1 because the large values for ovules do not correspond to the specified parameters of the table).

There were no valves with three, four, or five ovules as in *S. vitellina*. This specimen had the ovule index of 12–16 and the following distribution of valves with a different number of ovules: 6–52%, 7–34%; and 8–14%. The ovules were in the following arrangements on the two valves of the ovaries—6/6, 6/7, and 8/8. Such ovule distribution is typical for the “pure” *S. alba*, which grows in the natural areas in the south, specifically in the northern regions of Armenia, Turkey, and Iran [22]. *Salix alba* from the northern colder regions has valves with 3, 4, and 5 ovules, while in warmer or hotter climates, it has valves with a larger number of ovules ranging from 6 to 12 [1,20,22].

Thus, Specimen 15 represents a taxon different from *S. alba* f. *vitellina* by having broader leaves, a more pronounced orange–red color for the 1–3-year-old branchlets, and a greater ovule index of 12–16. Stott [41] listed ‘Vitellina Nova’ as a cultivar of *S. alba chermesina* and it is possible that the studied cultivar ‘Nova’ belongs to *S. alba* f. *chermesina*.

As mentioned above, in the protologue of *S. vitellina,* Linnaeus recorded characteristics of two different taxa (the first synonym one is dissimilar to the synonyms two and three). Of these, the second and third synonyms most likely correspond to f. *vitellina* and the first to f. *chermesina*. It is possible that Linnaeus observed these different taxa with bright yellow and bright orange–red stems and included them under the name of *S. vitellina.*

In summary, the female specimens of *S. alba* with yellow or red–orange stems were distinguished by their ovule number, which can be used to verify whether a specimen in question belongs to *S. vitellina* (ovule number 6–10) or *S. chermesina* (ovule number 12–16). For the taxonomic verification of the male plants (such as *S*. ‘Britzensis’, or *S. alba* (unranked)*vitellina britzensis* described by Spath in 1878 and noted to be a male plant (1930), hybridization studies can be conducted to verify plant identity when these plants can be used to pollinate *S. alba* with the known ovule index. After a female offspring reaches sexual maturity, its ovule number can be analyzed, and the ovule index of a male parent can be calculated to verify its identification.

## 4. Conclusions

The analysis of the floral morphology and ovule number clearly demonstrated that *S. vitelllina* belongs to *S. alba.* There was no evidence corroborating the hybrid origin of *S. vitelllina* as was recently proposed by Belyaeva et al. [19]. The genetic structure of *S. alba* is complex, as can be seen from its broad ovule index of 6–24 and the existence of many individuals with various proportions of ovaries with different ovule numbers [1,20,22]. The microscopic structure of the flowers of *S. vitelllina*, the ovule index of 6–10, and the ovule distribution (the presence of some valves with three ovules/ovaries and mostly with four and five ovules/ovaries) matched the subset of *S. alba*, previously designated as the *S. alba*-*fragilis*—Group.

Importantly, there was no aberration recorded for any of the studied specimens of *S. vitellina*, which would suggest its hybrid origin (androgynous or distorted flowers often occur in willow hybrids). At the same time, such abnormalities (the ovaries with three valves instead of two and androgynous catkins) were observed in the hybrids of *S. vitellina S.* ‘Oranzhevaya Tolstostvol’naya’ and ‘Pamyati Bazhova’.

Our recommendations: It is reasonable to designate the egg-yolk-colored willow as a form because the variation in the color of the stem is a minor or a “tertiary” morphological deviation. The representatives of *S. alba* with yellow stems originate at times in the native environments of the southern regions [15]. Shaburov, who studied *S. alba* in the Urals, also noted the sporadic occurrence of specimens of *S. alba* with yellow–orange stem coloration in southern locations [26]. The bright stem color is an inherited characteristic similar to the unusual form of willow stems such as those that are contorted or rounded (defined by the epithets “tortuosa” and “umbraculifera”) that are assigned the form status. Hence, the interpretation of *S. vitellina* as a form of *S. alba* is proposed here with the accepted name *S. alba* f. *vitellina* (L.) Wimm. (1866). Similarly, for *Salix alba* with orange–red stems, *S. alba* f. *chermesina* (Hartig) Rehder should be used as the accepted name.

Thus, our analysis confirmed that there are taxonomically different groups of *S. alba* with bright-colored stems—*S. alba* f. *vitellina* (ovule number 6–10) and *S. alba* f. *chermesina* (ovule number 12–16):

***S. alba f. vitellina*** (L.) Wimm., Salic. Eur.: 18 (1866).

**Basionym**: *Salix vitellina* L., Sp. Pl. 2: 1016. 1753 ≡ *S. alba* var. *vitellina* (L.) Stokes, Bot. Mat. Med. 506. 1812, ≡ *S. alba* subsp. *vitellina* (L.) Schübl. and G.Martens, Fl. Würtemberg: 630. 1834.

**Type**: Herb. Linn. No. 1158.13, female specimen (lectotype (Jarvis 2007)). The picture of the type specimen is available online (http://linnean-online.org/, accessed on 25 May 2023).

***S. alba* f. *chermesina*** (Hartig) Rehder, Bibliogr. Cult. Trees 75. 1949.

**Basionym**: *S. alba* var. *chermesina* Hartig, Vollst. Naturgesch. Culturpfl. Deutschl. 421. 1851.

In conclusion, it is recommended, for many taxonomically complex cases, therefore, that not only traditional morphological characteristics but the ovule number, microscopic floral characteristics, and molecular studies be provided for any taxonomic and nomenclatural innovations before the traditional use of the names in *Salix* be replaced to avoid any unnecessary disruptions.

## Figures and Tables

**Figure 1 plants-12-02610-f001:**
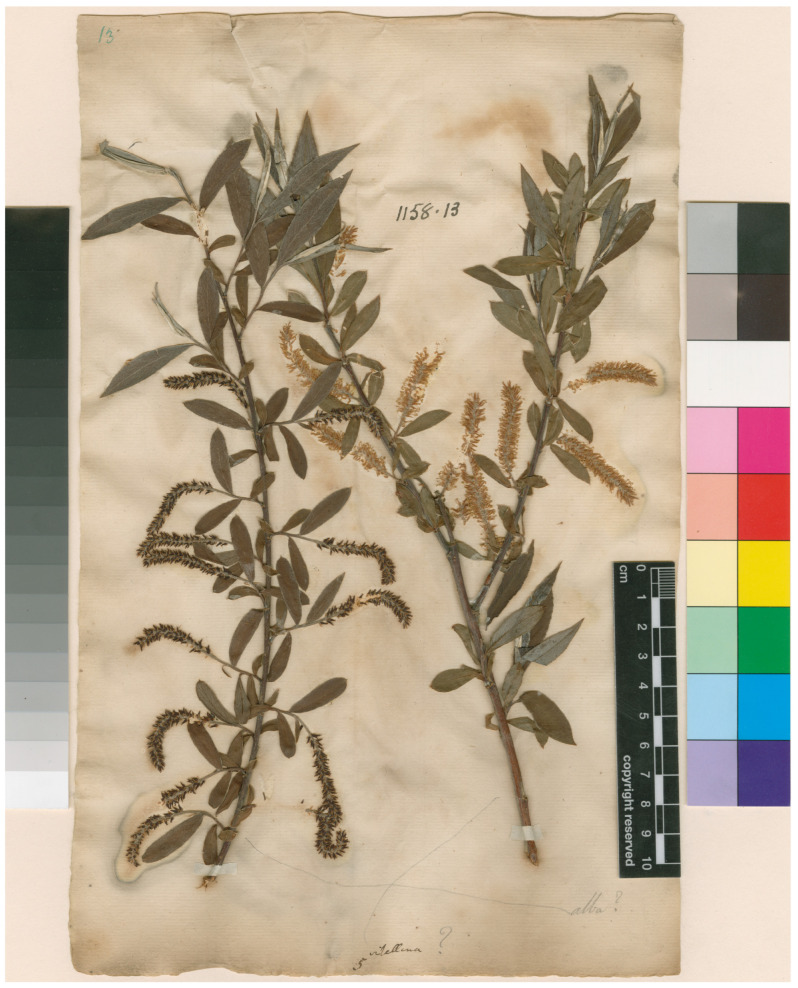
*Salix vitellina* Linn. No. 1158.13. The image of the type specimen is available online: http://linnean-online.org/, accessed on 25 May 2023.

**Figure 2 plants-12-02610-f002:**
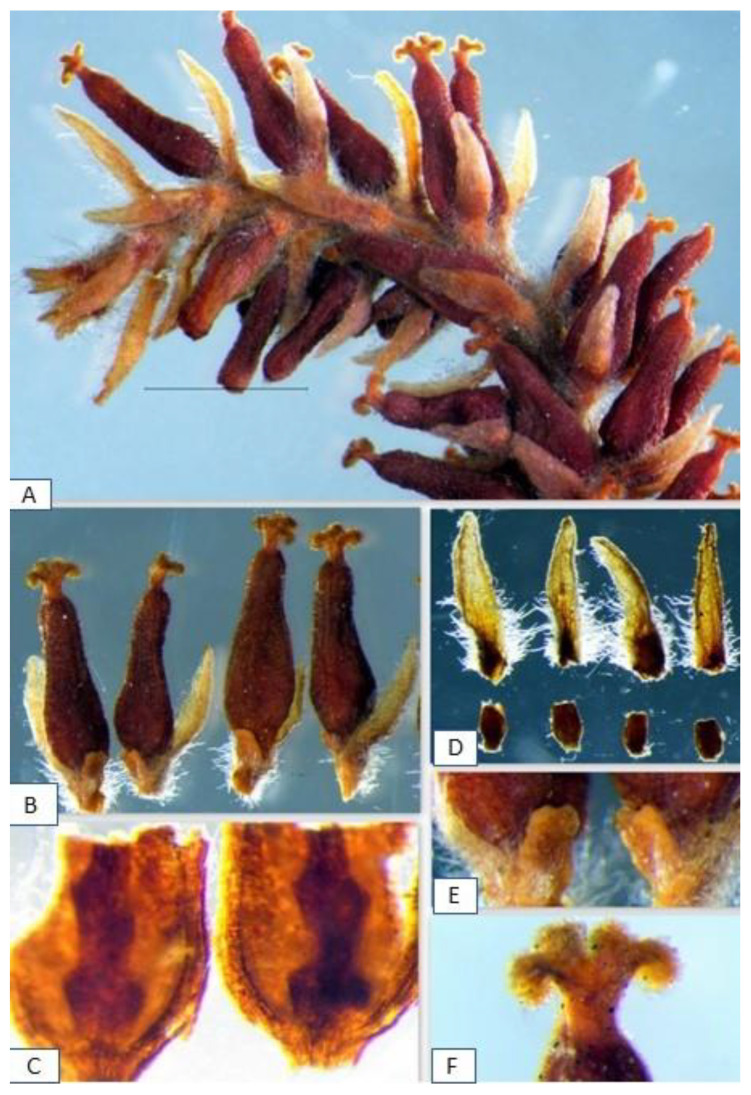
Linnaeus’s specimen of *S. vitelllina* LINN 1158.13: (**A**) a pistillate catkin with maturing capsules; (**B**) detached pistillate flowers where each flower consists of a single pistil with an ovary, floral bract, and adaxial nectary; (**C**) two valves of the ovary with traces of the four ovules on each valve; (**D**) dethatched floral bracts and nectaries; (**E**) adaxial nectaries; and (**F**) the stigma.

**Figure 3 plants-12-02610-f003:**
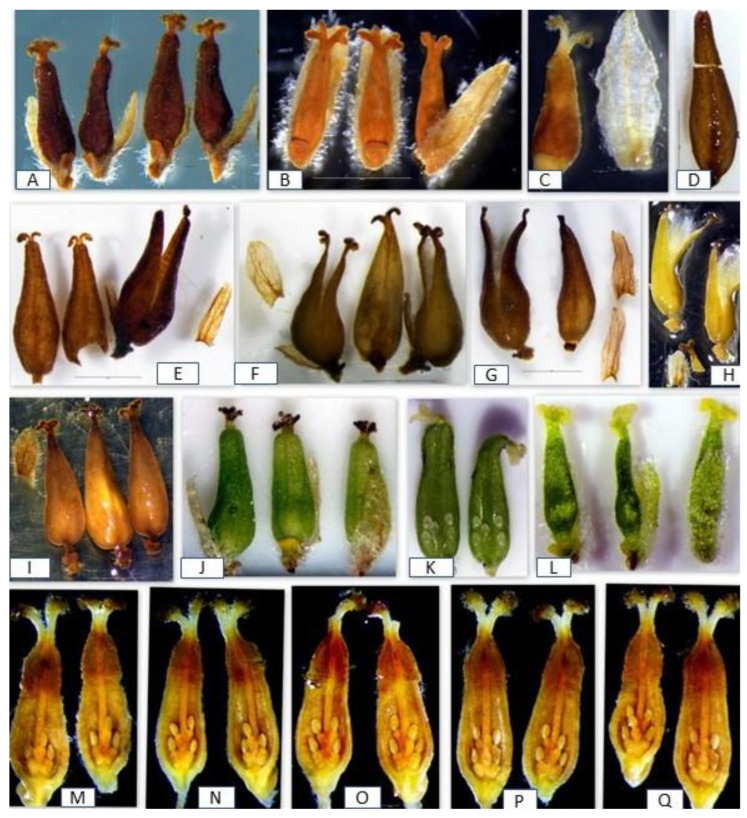
Detached pistillate flowers of the studied specimens: (**A**) specimen #1 (*S. vitellina* LINN 1158.13); (**B**) specimen #2; (**C**) specimen #3; (**D**) specimen #4; (**E**) specimen #7; (**F**) specimen #8; (**G**) specimen #9; (**H**) specimen #10; (**I**) specimen #12; (**J**) specimen #13; (**K**) specimen #13 (two valves of the capsule with ovules); (**L**) specimen #14; (**M–O**) specimen #3 with all possible ovule numbers and distributions; (**M**) an ovary with 6 ovules (3/3 per valve); (**N**) an ovary with 7 ovules (3/4 on each valve); (**O**) an ovary with 8 ovules (4/4 per valve); (**P**) an ovary with 9 ovules (4/5 per valve); and (**Q**) an ovary with 10 ovules (5/5 per valve).

**Figure 4 plants-12-02610-f004:**
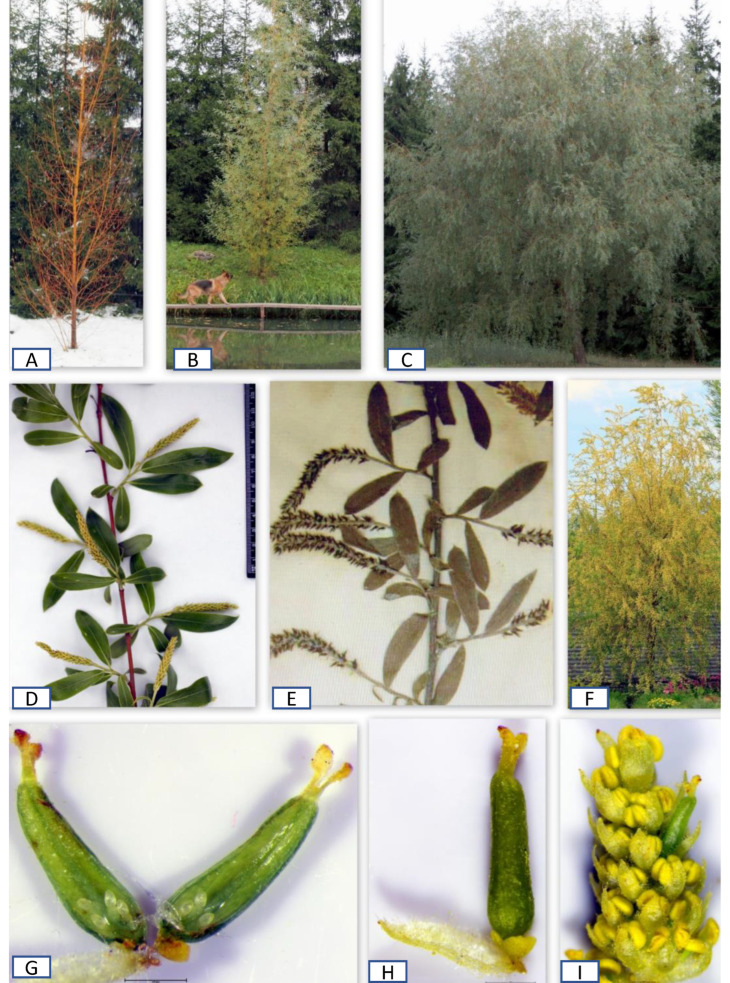
(**A**,**B**) An elongated crown of *S*. ‘Oranzhevaya Tolstostvol’naya’ in winter and summer during the first few years; (**C**) the broad slightly weeping crown of *S*. ‘Oranzhevaya Tolstostvol’naya’ after 10 years; (**D**) a branch fragment of ‘Oranzhevaya Tolstostvol’naya’; (**E**) a branch fragment of LINN 1158.13; (**F**–**I**) *S*. ‘Pamyati Bazhova’; (**F**) abundant flowering in spring; (**G**) the two valves of the ovary with four and three ovules on each of the two valves; (**H**) a female flower discovered in the male catkin; and (**I**) an androgynous catkin with one female flower.

**Figure 5 plants-12-02610-f005:**
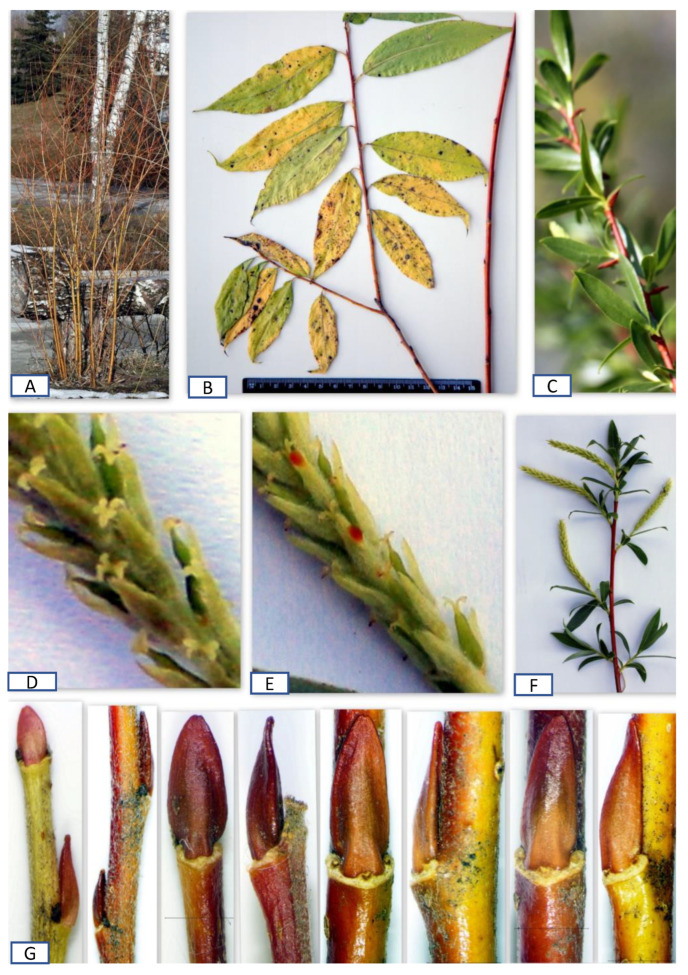
*S. alba* f. *chermesina* ‘Nova’: (**A**) bright colored stems in early spring; (**B**) a branchlet with ovate-lanceolate leaves at the end of the growing season; (**C**) a branchlet with attached bud scales in spring; (**D**,**E**) female catkins; (**F**) a flowering branchlet; and (**G**) branchlets with buds in late autumn. The dry stipules are present as dark spots on the sides of the buds in the two photos on the left.

**Table 1 plants-12-02610-t001:** The ovule data for *S. vitellina* and related specimens. The presented data include the percentage of valves and ovaries with different numbers of ovules, and the ovule indices (the minimum and maximum number of ovules per ovary).

No.	Specimen	Ovule Number								Ovule Index	No. of Ovaries	Proposed Taxon
		3	4	5	6	7	8	9	10			
		Percentage of Valves with Each Ovule Number			Percentage of Ovaries with Each Ovule Number							
*S. vitellina*												
1	*S. vitellina* LINN 1158.13	31.5	37	31.5	11	28	26	20	15	6–10	54	*S. alba* f. *vitellina*
2	*S. vitellina* (Frischer/*S. fragilis*–Anderson) Germania. LE 838	28	43	29	13	28	16	32	11	6–10	63	*S. alba* f. *vitellina*
3	*S. alba* var. *vitellina* (Alfred Rehder) No. 8460. Washtenaw County, Michigan, US. June 1938. Escaped or planted. A	13	55	32	2	16	39	29	14	6–10	58	*S. alba* f. *vitellina*
4	*S. alba* var. *vitellina * (Emil Hausser/*S. alba* L.–I.V.Belyaeva) north-east France, Barr, diluvium, 200 m. 29.04 and 28.05,1885. K	25	50	25	-	50	-	50	-	7–9	2	*S. alba* f. *vitellina*
Additional specimens												
5	*S. alba* L. *× S. fragilis* L. (A.K.Skvortsov) Arboretum KazNIILKH, Shchuchinsk, Kokchetav region, Kazakhstan. MHA	10	53	27	3	11	28	44	14	6–10	31	*S. alba* (*S. alba-fragilis—Group*)
6	*S*. *fragilis* L. *× S*. *alba* Wimm. (E.T.Malyutina) Kuznetsk district, between the village of Dvoriki and the village of Mordovsky Kachim, Penza region, Russia. MHA	13	84	3	2	23	71	3	1	6–10	91	*S. alba* (*S. alba-fragilis—Group*)
7	*S. alba* L. (A.K.Skvortsov) Leninogorsk, Zap. Altai, Russia. MHA	27	46	27	15	9	39	33	4	6–10	33	*S. alba* (*S. alba-fragilis—Group*)
8	*S. alba* L., (A.K.Skvortsov) Valley of the river Khoper near Balashov, Saratov region, Russia. MHA	25	59	16	8	32	34	23	3	6–10	65	*S. alba* (*S. alba-fragilis—Group*)
9	*S. alba* L. (A.K.Skvortsov) Floodplain of the Oka, Serpukhov district, near the village of Luzhki, Moscow region, Russia. MHA	24	33	43	3	17	38	25	17	6–10	63	*S. alba* (*S. alba-fragilis—Group*)
10	*S. alba* var. *caerulea,* (A.B.Jackson/*S*. *alba* L.- I.V.Belyaeva) Near the water near Longmore, Middlesex, England. K	36	27	37	18	14	25	32	11	6–10	28	*S. alba * (*S. alba-fragilis—Group*)
11	*S*. *fragilis* L. (E.T.Malyutina) Kiryushkino village, Buguruslan district, Orenburg region, Russia. MHA	34	41	25	22	21	23	21	13	6–10	71	*S. alba * (*S. alba-fragilis—Group*)
12	*S*. *fragilis* L. (E.T.Malyutina) Tyukhmenevo village, Kuznetsk district, Penza region, Russia. MHA	23	67	10	11	23	46	18	2	6–10	73	*S. alba * (*S. alba-fragilis—Group*)
13	*S*. ‘Oranzhevaya Tolstostvol’naya’ (A.M.Marchenko) NA-0102510, Russian Park Water Gardens. Pushkino Moscow region, Russia. 15 May 2017. NA	39	34	27	14	24	38	18	6	6–10	139	*S. alba* f. *vitellina × S. fragilis- alba —Group*
14	*S. alba* L. (V.I.Shaburov; late weeping form) Krasnoufim region Sverdlovsk Oblast iso SVER	50	48	2	27	42	27	4	0	6–9	146	*S. fragilis* var. *alba—Group*

## Data Availability

Data available in a publicly accessible repository at the Center for Open Science’s Open Science Framework (https://osf.io/kdqm5/, accessed on 25 May 2023).

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
