# Peer review of "Notes on the Taxonomy of Salix vitellina (Salicaceae)"

_plants, 2023, doi:10.3390/plants12142610_

Round 1
Reviewer 1 Report
The authors describe in their manuscript "Notes on the taxonomy of Salix vitellina (Salicaceae)" the taxonomomic status of the named species and discuss it based on ovule characteristics. Overall it is a single-taxon study focussing on one single trait. However, the general outline describing and analysing this trait is sound and leads to the finding to treat this specific taxon as a "form". Although the amount of molecular studies are rising, sound taxonomical studies on non-molecular traits are still of value.
The introduction directly starts with taxonomic details. In my opinion at least 2-3 sentences on willows in general and the taxonomic difficulties would help the majority of readers (who might not be familiar with genus Salix) to place this study. Also, the information on the necessity to discuss this specific taxon should be placed in front of the taxonomic details.
Since the discussion includes S. fragilis (former euxina), S. alba and S. vitellina, all three species should be also described/mentioned in the introduction and the existing knowledge about the relationships of all three species should be briefly listed.
The ovule method is mentioned in the introduction - this method was used by the authors. However, the authors exclusively cite themselves. Are there other studies using this method (maybe in other plant taxa) for taxonomical studies?
The discussion is quite long and detailed on single individuals. In this case of a taxonomic study this might be neccessary, however, maybe it is possible to shorten it a bit and focus on the main outcome. The discussion would also benefit from some more general comparisons not only focussing on this specific Salix form but, however, to also open a broader view on this kind of studies for taxonomically challenging groups.
The conclusion of the authors is, to treat S. vittelina as a form of S. alba. Therefore, the introduction should prepare the reader to this outcome. Are there more forms of S. alba? Is this a common way to deal with different morphological forms in genus Salix? Some more general information would be good for non-Salix specialists.
Minors
Species names should be italic throughout the manuscript
p1, L36 - double space
p1, L44 - POWO - Write it out when first mentioned
p2, L 49 "a" missing before "high degree"
p2, 79 double space - there are more such formal minors in the manuscript, please check (I will not mention all of them).
p4, 114: Fig. Capture "....loral bracts" - should be floral bracts
p6. 128 - "The genetic structure..." - this sentence should be part of the discussion (interpreting the results). In any case, some molecular studies could be mentioned
Discussion - the names of different ranks and individuals are displayed in different ways. Maybe you find a concise way to use capital and low letters regarding the taxonomic rank of the species/taxon (e.g. S. ‘Oranzhevaya Tol-237 stostvol'naya’ ; ‘Pamyati Bazhova’; "S. alba 'Nova'; ...)
"specimen" should be written with low letter "s" (except begin of sentence)
p.12, L 296 - the line break is somewhat strange, please correct
Overall the manuscript is well written. I only detected rather minor issues.
Reviewer 2 Report
The manuscript presents a morphological study on Salix vitellina to clarify its taxonomy and nomenclature. The paper is valuable because it corrects some erroneous classifications and nomenclatural confusions that were caused by some recent papers of other authors (Belyaeva). The paper is important and should be published. Some revisions of the manuscript should bring more clarity and make their conclusions more convincing.
Introduction: Lines 43-44: after S. ×fragilis f. vitellina (L.) 43 I.V.Belyaeva please give the year of publication (as for the other synonyms), and briefly explain what this author meant here –S. xfragilis was taken erroneously as binomial for the hybrid formula S. alba x fragilis (according to Marchenko & Kuzovkina 2022). In previous papers the authors already clarified that Belyaeva’s view on the nomenclature of S. fragilis (and the application of the name S. euxina to the crack willow) was wrong and one should return to the classical nomenclature of S. alba L. and S. fragilis L. and their hybrid S. xrubens. But this should be explained here for the reader who is not necessarily aware of the previous papers, and the confusion of names caused by the Belyaeva papers. The POWO citation can be omitted – it is just a database of taxa (with many mistakes and often criticized in other genera – best simply ignored), it is no primary taxonomic work which would be relevant here. I would rather recommend to cite Meikle 1984 who gives very detailed descriptions of varieties and hybrids of this group.
Material and methods: the ovule number appears to be a good character, but it is restricted to females and a certain developmental stage. One would not base species delimitation on a single character. It would be more convincing if the authors would use all diagnostic characters of S. alba versus S. fragilis. Beside the characters listed here, the authors could check the floral bracts: in S. alba there are curled hairs on the base of the bract, and the tip is glabrous, whereas pure S. fragilis has straight long hairs on the tip of the bract down to the base (see (Hörandl et al., 2012), p. 62). The hybrids S. alba x fragilis usually combine both hair types. The type of S. vitellina has obviously bracts like typical S. alba, as shown in Figure 2 of the manuscript. Also, the classical leaf characters could be used if compared at the same stage (late flowering time): typical S. alba is hairy beneath, typical S. fragilis is glabrous. S. alba is glabrescent during the season which often makes difficulties to discriminate it from the hybrid or from S. fragilis, but it is a good character in spring. S. alba has hairy twigs (2nd year), S. fragilis is glabrous. S. vitellina leaves are indeed more often glabrous, as mentioned by Belyaeva 2018, but to my field experience this could be due to frequent cutting of branches of these ornamental trees, so that the tree produces long and elastic, rod-like twigs that were formerly used for making baskets. Nowadays this is often just done for an ornamental habit. This habit is shown e.g. in Belyaeva 2018 in the photo in Figure 2. Cutting has an influence on leaf morphology, especially on indumentum.
From the methodical side it would be more convincing to score all characters and tabulate them, and it would be recommended to conduct a multivariate statistical analysis of samples (principal coordinate and principal component analyses) instead of using just descriptions of a single character, which is further only available on the female plants.
Discussion:
In the introduction/discussion the authors should also mention the genetic studies on S. alba and S. fragilis and their hybrids which clearly showed separated gene pools, e.g., (Barcaccia et al., 2014) and many previous papers cited therein.
One could mention in the discussion that (Neumann, 1981) p. 72, under S. alba subsp. vitellina, wrote that the taxon has a possible native origin in the Balcan-Peninsula (var. vitellina), because it occurs in Croatia along riversides like a wild plant. In Central Europe the taxon is only cultivated.
What is not mentioned in Belyaeva 2018: S. alba var. vitellina x S. babylonica (S. x sepulcralis nothovar. chrysocoma) is a very commonly cultivated weeping willow in Europe. (Meikle, 1984) gives nice descriptions how to discriminate it from S. x pendulina forms (= S. fragilis x babylonica). Belyaeva 2018 does not mention S. x sepulcralis and describes only S. x pendulina as weeping willows. I have some suspect that here some misidentifications of weeping willows played a role in the descriptions.
Conclusions:
I recommend to present a formal taxonomic conclusion and revised lists of synonyms on the taxon, as it was presented in Belyaeva 2018. This is then clearer and easy-to-grasp for taxonomists and flora-writers, who might not want to read all details of the text.
Minor points:
Line 9: “… a form of a hybrid S. xfragilis”. This statement is a bit confusing because the authors do not accept the interpretation of S. xfragilis sensu Belvaeva, but rather regard S. fragilis a species (based on previous papers) and the hybrid S. alba x fragilis should be named S. xrubens. Better use your accepted names (which are in common use) in the abstract. See also comment on lines 43-44.
Line 72: Prühonice– correct name of this village is Průhonice
Recommended literature:
Barcaccia G, Meneghetti S, Lucchin M, de Jong H. 2014. Genetic segregation and genomic hybridization patterns support an allotetraploid structure and disomic inheritance for Salix species. Diversity, 6: 633-651. Doi: 10.3390/d6040633
Hörandl E, Florineth F, Hadacek F. 2012. Weiden in Österreich und angrenzenden Gebieten [Willows in Austria and adjacent regions]. Vienna: University of Agriculture, Vienna. Doi:
Meikle RD. 1984. Willows and poplars of Great Britain and Ireland. London: Botanical Society of the British Islands. Doi:
Neumann A. 1981. Die mitteleuropäischen Salix-Arten. Mitt. Forstl. Bundes-Versuchsanst. Wien, 134: 1-152. Doi:
